# Supply–Demand Evaluation of Green Stormwater Infrastructure (GSI) Based on the Model of Coupling Coordination

**DOI:** 10.3390/ijerph192214742

**Published:** 2022-11-09

**Authors:** Mo Wang, Furong Chen, Dongqing Zhang, Qiuyi Rao, Jianjun Li, Soon Keat Tan

**Affiliations:** 1College of Architecture and Urban Planning, Guangzhou University, Guangzhou 510006, China; 2Guangdong Provincial Key Laboratory of Petrochemical Pollution Processes and Control, School of Environmental Science and Engineering, Guangdong University of Petrochemical Technology, Maoming 525000, China; 3Architectural Design & Research Institute, Guangzhou University, Guangzhou 510499, China; 4School of Civil and Environmental Engineering, Nanyang Technological University, Singapore 639798, Singapore

**Keywords:** green stormwater infrastructure, supply–demand, coupling coordination model, sponge city

## Abstract

The rational spatial allocation of Green Stormwater Infrastructure (GSI), which is an alternative land development approach for managing stormwater close to the source, exerts a crucial effect on coordinating urban development and hydrological sustainability. The balance between the supply and demand of urban facilities has been an influential standard for determining the rationality of this allocation. However, at this stage, research on evaluating planning from the perspective of supply–demand in GSI is still limited. This study proposed an evaluation method for assessing supply–demand levels in GSIs in Guangzhou, China, using the coupling coordination model consisting of Coupling Degree (CD) and Coupling Coordination Degree (CCD). Furthermore, the spatial distributions of supply–demand balance and resource mismatch were identified. The results indicated that the supply and demand levels of GSI exhibited significant spatial differences in distribution, with most streets being in short supply. The GSI exhibited a high CD value of 0.575 and a poor CCD value of 0.328, implying a significant imbalance in facility allocation. A lot of newly planned facilities failed to effectively cover the streets in need of improvement, so it became essential to adjust the planning scheme. The findings of this study can facilitate the decision-makers in assessing the supply–demand levels in GSI and provide a reference of facility allocation for the sustainable construction of Sponge City.

## 1. Introduction

Fast population growth, rapid urbanization, and climate change have caused disturbance of the natural landscape with incremental impervious surfaces, exerting a profound impact on hydrological process [1]. The change in land use has resulted in increased velocity and volume of surface water runoff, decreased retention time, and deterioration of water quality [2,3,4]. In general, urban runoff management’s traditional goal has been to control flooding by storing stormwater in storm sewers that discharge in surface bodies directly [5]. The gray infrastructures have been the primary approach for conveying and storing stormwater–either separately or in combination with sewage flows–during precipitation events [6,7,8]. However, the extreme precipitation resulting from urbanization has posed critical challenges to urban drainage infrastructure. These gray infrastructure systems are vulnerable to adapting to urban environments with high density. Especially during heavy and continuous rainfall, the urban drainage system is unable to discharge rainwater smoothly, resulting in a large amount of surface runoff trapped in the streets, which leads to urban flooding [9,10,11].

Green Stormwater Infrastructure (GSI), as an innovative stormwater management approach and nature-based solution, has been highly recommended for reducing stormwater runoff, improving water quality, and providing ecosystem sustainability [12,13,14,15]. The main purpose of GSI is to reduce the impact of development on water-related problems through the use of GSI practices such as bioretention, green roofs, grass swales, and permeable pavements that infiltrate, evaporate, and use stormwater on site where it falls [16]. In recent years, more research has been carried out on GSI practices, and the use of this practice has shown magnificent benefits in stormwater management owing to its ability to reduce runoff and soil loss and improve water quality [17,18,19]. Since 2015, China has been advocating for the Program of Sponge City Construction, which refers to the establishment of GSIs in urban catchments to ensure hydrological safety and ecological water quality for the city. To date, plenty of GSIs have been constructed successively in China, with a total investment of 80 billion CNY [20]. In 2021, twenty cities, including Guangzhou and Hangzhou, were selected as the demonstration of Sponge City thanks to their excellent construction performances [21].

GSI has emerged as an important tool for improving the urban sustainability, climate change adaptation, and resilience of urban hydrology, given its capacity to provide ecosystem services, such as retaining stormwater runoff [22,23], non-point source pollution mitigation [24,25], air purification [26], heat-island effect alleviation [27], and city landscape shaping [28]. However, the construction of a sponge city has been quite controversial due to the remaining urban flooding at various levels [29]. Among the 351 cities investigated in China, 62% suffered from urban floods, and 137 cities suffered from severe floods more than three times a year [30]. The main reasons responsible for this phenomenon are the imbalance in the distribution of supply–demand for GSI facilities [31], lack of effective maintenance [32], and overestimation of GSI facility performance [33]. Among them, the imbalance in the distribution of the supply–demand of GSI facilities has been particularly regarded as the most critical issue [34]. de Manuel et al. [35] revealed that 35% of communities in Bilbao possess a GSI supply that is far from sufficient for meeting the local demand for ecosystem services, such as urban flooding control.

Furthermore, when encountering unprecedented climate change and urbanization, it is more important to strengthen the response and resilience to the risk disturbances of urban infrastructure. The benefits of ecosystem services for climate adaptation depend largely on where and how stormwater infrastructure is locally supplied [36]. The degree of supply of stormwater facilities has been generally regarded as the criterion for adapting to rainstorms and alleviating urban flooding [37]. The commonly accepted assessment method is to evaluate the hydrological performance of stormwater facilities under rainfall conditions by applying hydrological and hydraulic models so as to identify the extent of urban flooding. Li et al. [38] proposed a GSI supply evaluation method based on the flooding accumulation model and found that the supply of facilities in central-urban areas was significantly lower than that in suburban areas. Mei et al. [39] integrated the assessment models of the Storm Water Management Model and Life Cycle Assessment to evaluate the hydrological performance of GSIs and identified the spatial distribution differences of GSIs in supply. Joycee et al. [40] employed a multi-scale methodology based on the hydrological model to quantitatively analyze the GSI supply degree in coastal areas of Florida, USA. However, the hydrological and hydraulic models relied too much on catchment geomorphological data and monitoring data, which suffered significantly from the complexity of operation [41,42]. Therefore, it is difficult to quickly and effectively evaluate the supply level of GSIs based on the application of these methods [43].

In addition to assessing the supply level in terms of quantity, area, or performance, the matching degree of supply and demand of GSI is also an essential index for the rationality of facilities’ spatial allocation [34]. The evaluation of the supply and demand levels of facilities should generally possess the following characteristics: (1) the ability to identify the interaction between the supply and demand of various facilities, and (2) the provision of scientific methods to evaluate the supply and demand levels under representative scenarios. For the spatial allocation of urban land uses and facilities, there have been numerous measurement studies evaluating the levels of supply and demand in services. Meerow and Newell [44] proposed a simplified and efficient method for evaluating facility supply, seeking a rigorous urban hydrological process rather than directly analyzing the spatial distribution of natural basement and existing facilities. Their results indicated that an obvious mismatch between supply and demand for GSI was observed in Detroit, USA. Ramyer et al. [45] developed an evaluation model with the matrix combination of social demand and ecosystem services of GSI and identified Tehran as the area with the severest supply–demand mismatch. Heckert and Rosan [46] characterized the supply level via built-up environmental factors and the demand level via socio-economic variables. Taking Philadelphia as an example, the authors evaluated the supply/demand level and distribution of the relevant GSI and found that the top 20% of the most needed communities were not fully covered by the GSI projects. The above studies mainly focused on the numerical difference between supply and demand systems, directly reflecting the rationality of spatial resource allocation. However, Zhang et al. [22] found that controlling the proportion of GSI facilities within the watershed of Guangzhou to 24.4% achieved the maximum urban flood control effect. Thus, the supply–demand balance of GSI cannot be measured only by the numerical difference between supply and demand because there exists an acceptable threshold value for the supply and demand system of GSI.

The coupling coordination model consisting of Coupling Degree (CD) and Coupling Coordination Degree (CCD) is a method used to determine a threshold to identify whether multiple systems are in balance [47,48,49]. Sun et al. [50] identified areas where the current and future supply and demand of ecosystem services in China are balanced based on the coupled coordination model, using the coupling condition that the CD exceeds 0.6 and the CCD value exceeds 0.5. The CD and CCD that do not exceed this predetermined value indicate a state of the ecosystem service in undersupply or supply exceeding demand. Since the coupling coordination model can be used to describe the levels of resource allocation subject to supply and demand objectively and concisely, it has already been broadly applied to evaluating the synergy of ecosystem services [51], environmental justice [52], and land-use optimization [28]. The above studies have discussed the planning or adjustment policies of specific implementation schemes mainly at the national and regional level [7,53], but rarely on the scale of urban built-up areas. As a scientific and functional method, the coupling coordination model exhibited the potential to examine the coordination statuses of GSI planning and construction based on urban street units and weigh the supply/demand levels of GSIs [54]. However, to date, few studies have been carried out to assess the supply–demand relation and examine the potential spatial mismatches in the supply and demand of urban ecosystem services with GSI facilities using a coupling coordination model. Within this context, the main objectives of this study are to: (i) propose a GSIs’ coupling coordination measurement model applicable to urban street units; (ii) assess the supply/demand levels of GSIs; and (iii) identify the spatial distributions of supply/demand balance and resource mismatch. This method facilitates the effective examination of the rationality of GSI in spatial allocation and provides a reference for the planning and regulatory policies of Sponge City constructions.

## 2. Materials and Methods

Within the research framework (Figure 1), the GSI supply and demand levels of urban street units were specifically examined. Then, the CDs and CCDs of GSIs were quantified in the spatial distribution, so as to evaluate the spatial distributions of the GSI supply–demand balance and identify the mismatch between supply and demand system.

### 2.1. Case Study

Guangzhou has abundant rainfall, with an annual precipitation of 1800 mm on average. However, affected by the East Asian monsoon, its rainfall distribution has been uneven throughout the year [28]. The central-urban districts of Guangzhou (Figure 2) were selected as the case study area in the present study, including Yuexiu, Liwan, Haizhu, Tianhe, Huangpu, and Baiyun Districts, with a population of over 10 million residents. A decade ago, Guangzhou was one of the largest coastal cities with the severest urban flooding all over the world [55]. In recent years, due to the efficient implementation of large-scale GSIs construction. Guangzhou has been regarded as a role model for Sponge City in China. However, urban flooding still occurs in this city from time to time [56].

### 2.2. Data Sourcing and Processing

The sourced data include urban flooding events and relevant spatial variables associated with the supply–demand relationship of GSI, as shown in Table 1. The cumulative number of urban flooding events was adopted as an index of the demand degree of the GSI [44]. Therefore, the observed urban flooding events were transformed into point data based on geographical coordinates. The more urban flooding occurs on the street, the higher the level of demand. Due to the limitations of access to the observed data, 15 streets were excluded from the present study.

The vegetation coverage, parks, water bodies, playgrounds, and impervious surfaces were selected as indexes for assessing the supply levels of GSIs based on the relevant literature [36]. Vegetation coverage could serve as a natural landscape with important functions of the retention and infiltration of surface runoff within urban catchments. Nature-based solutions, parks, and water bodies might be regarded as facilities for effectively retaining runoff. The playgrounds could be applied as detention facilities for surface runoffs during rainstorms, which have good flexibility in function. Impervious surface is a reverse index of GSI supply degree, which usually reflects the occurrence possibility of urban flooding [57]. 

Since the original indexes and the dynamic relative quantities usually have different dimensions and orders of magnitude, it is necessary to standardize the indexes to exclude the influence of these factors. The numerical value of streets with vegetation coverage over 36% was set to 1 [58]. Streets with coverage ranging from 0 to 36% were standardized accordingly. The higher the proportion of impervious ground, the lower the supply of GSI. Compared with other indicators, this was a “negative index” and required special consideration. In this study, the dispersion normalization method was used to standardize the original indexes to the range of [0,1] [59], with the calculation formula (1) described below:(1)$${\hat{x}}_{ij}=\left\{\begin{array}{c} \frac{x_{ij}-\min(x_{i})}{\max(x_{j})-\min(x_{j})}\begin{array}{cc} \;\;\;\;\mathrm{Positive}\;\mathrm{indictors} & \end{array}\\ \frac{\min(x_{j})-x_{ij}}{\max(x_{j})-\min(x_{j})}\begin{array}{cc} \;\;\;\;\mathrm{Negitive}\;\mathrm{indictors} & \end{array} \end{array}\right.\;$$
where ${\hat{x}}_{ij}$ is the value after standardization, $x_{ij}$ is the value of index *j* in street i, and $\min(x_{j})$ and $\max(x_{j})$ are respectively the minimum and maximum values of index *j* of street i.

To determine the indicator weight [51,59], the proportion of the *j*th indicator value in street i was calculated as follows:(2)$$w_{ij}=\frac{{\hat{x}}_{ij}}{\sum \begin{array}{c} m\\ i=1 \end{array}{\hat{x}}_{ij}}$$
where $w_{j}$ represents the proportion of the jth indicator value in street *i*.

The information entropy of each indicator and the redundancy degree of the information entropy were determined as follows:(3)$$e_{j}=-\frac{1}{\ln m}\sum_{i=1}^{m}(w_{ij}\times \ln w_{ij})$$
(4)$$d_{j}=1-e_{j}\;$$
where $e_{j}$ is defined as the information entropy of the jth indicator; $0\leq e_{j}\leq 1$, $d_{j}$ is the redundancy degree of the information entropy.

The weight was calculated as follows:(5)$$w_{j}=\frac{d_{j}}{\sum \begin{array}{c} m\\ j=1 \end{array}d_{j}}$$
where $w_{j}$ is the weight of the jth indicator.

The level of supply of GSI in the central city was calculated based on the dimensionless normalized values and the weights of each index:(6)$$S(x)=\sum_{i=1}^{n}w_{i}x_{i}$$
where *S*(x) are the comprehensive level of supply, $x_{i}$ represents the normalized index values of the GSI supply level, and $w_{i}$ denotes the corresponding weights.

In general, any statistical series may have certain natural turning points and characteristic points that can be used to divide study subjects into groups of a similar nature. In contrast, Natural Breaks is a statistical method for grading and classifying indices according to their statistical distribution, and it can maximize the difference between classes. Therefore, to classify the levels of supply–demand, the Natural Break in ArcGIS 10.5 was used. The above indexes were divided into three levels: high, moderate, and low [57].

### 2.3. Analysis of the Relationship between GSI Supply and Demand

The coupling coordination model has two indexes: Coupling Degree (CD) and Coupling Coordination Degree (CCD). The coupling levels and the coupling coordination levels were set according to previous studies [50,60], as shown in Table 2. CD was used to describe the degree of influence of the interaction between subsystems [48,49,50,51]. Therefore, the numeric value of CD can effectively reflect the intensity of the interaction between GSI supply and demand subsystems, with the corresponding calculation in Formula (7):(7)$$CD={\left\{\frac{S(x)D(y)}{{\left[\frac{S(x)+D(y)}{2}\right]}^{2}}\right\}}^{2}$$
where CD is the coupling index, with the value range of [0,1]; *S*(*x*) denotes the comprehensive index of the supply subsystem, which is obtained by weighting the standardized supply-layer factors via the entropy weight method; *D*(*y*) presents the numeric value of the demand, measured by the frequency of urban flooding. The larger the index, the higher the coupling degree between GSI supply and demand in the studied region [47,48].

CCD can be used to identify the trend of the subsystem from disorder to order in order to judge the threshold of the rationality of facility allocation [49,54]. The coordination level of GSIs’ supply–demand relationship can be identified through CCD, which enables the optimization direction of facility allocation to be accurately determined. The calculation of the CCD is as follows:(8)$$CCD=\sqrt{CD\times T}$$
where CCD is the coupling coordination index. The larger the *CCD* value, the better the surface coordination [50]. T is the coefficient reflecting the comprehensive levels of supply and demand of GSIs, with its calculation formula as below:(9)T=αS(x)+βD(y)
where α and β represent the contributions (weights) of supply and demand subsystems, respectively, and the two values are both set to 0.5 without specific conditions.

## 3. Results

### 3.1. GSI Supply and Demand Levels

In the present study, the coupling coordination model was employed to identify the spatial differentiation degree of supply and demand in the GSI. If the degree of supply was higher than public demand, it indicated that the government’s investment in GSI had problems, resulting in a waste of resources; when the public demand was higher than the government supply, it means that citizen satisfaction was mainly based on a good ecological environment, and the government supply was insufficient; when the two were equal, it showed that the supply and demand for GSI were in balance [54,61]. 

The demand index for the central urban districts of Guangzhou was calculated as 0.737, while its supply index was only as low as 0.264. The overall tendency of the demand-supply correlation seemed to be typical “supply over demand,” and the gap was significant (Table 3). Despite the recent large-scale construction of GSI, there remains a huge demand for GSI supply services in general. Among them, Tianhe District showed the largest gap between supply and demand, with a value of 0.583, while Liwan District exhibited the smallest gap between supply and demand, with an index value of 0.387.

In terms of the supply level, the obvious spatial differences in the distribution of GSIs were observed (Figure 3a). Tianhe District had the largest number of low-supply streets, followed by Baiyun and Huangpu, while Liwan District had the least. The high-supply streets were mainly concentrated in the middle-north of Huangpu District, the east of Baiyun District, and the middle-east of Haizhu District, suggesting that there was a significant difference between the GSI supply levels of Baiyun and Huangpu Districts. In general, the streets with a high supply were centered around or adjacent to open spaces, such as urban parks, scenic spots, and wetlands. On the other hand, Tianhe District had the smallest number of streets with a high supply, accounting for only 10% of the total. This finding was attributed to the fact that the Tianhe District had plenty of buildings and roads with high density, leading to a substantial impervious surface proportion.

GSI also presents obvious spatial differences on the demand side. The number of streets with high demand was significantly higher than those with low demand. As shown in Figure 3b, the number of high-demand streets in the Tianhe and Baiyun Districts accounts for more than 50% of the total number, while the value is 36% in Huangpu and 21% in Liwan, respectively. Most of the high-demand streets were characterized by the scarcity of green spaces. The exception was Tonghe Street in Baiyun District, with a concentrated large area of green space, which still exhibited a high demand. This finding may be attributed to the inability to urban flooding suppression once the green space exceeds a certain threshold.

### 3.2. Coupling and Coordination Degrees of GSI Supply and Demand

In the present study, the coupling coordination model was employed to identify the spatial differentiation degree of supply and demand in the GSI. The results indicated that the coupling degree of the GSI supply/demand was classified as moderate (0.575), while their coordination level seemed to be low (0.328). The coupling degree index ranged from 0.169 to 0.998, with substantial fluctuations. Sha Mian Street in Liwan District exhibited the highest coupling degree, while Jinhua Street, located in the same district, exhibited the lowest one. According to the numerical magnitude of the coupling, five levels are classified, as presented in Table 2. Extreme and serious decoupling seemed to occupy the majority of streets (Figure 4), which were primarily distributed in the middle-south of Tianhe, the south of Huangpu, the middle of Haizhu, and the northwest of Baiyun. Primary coupling streets, with the second largest number, were mainly distributed in the north of Huangpu, central Liwan, and the eastern part of Yuexiu (Figure 4c). These streets generally exhibit moderate demand compared to other areas (Figure 3c), and have a higher level of supply compared to other streets due to the large green areas in these streets. Specifically, the north of Huangpu was covered with plenty of natural green spaces, and the primary coupling streets in Liwan and Yuexiu Districts were generally occupied with a lot of park yards and many nature-oriented landscapes. Thus, the coupling relationship between the supply and demand of the street in these regions was verified to be benign. By contrast, the coupling degree of most streets in the Tianhe District was low, 84% of which were seriously decoupled.

In view of the different development speeds of different regions, there were obvious differences in the coupling and coordination degrees between supply and demand in GSI in various regions. According to the CCD analysis, the quality of supply–demand coupling in GSI in Guangzhou was not ideal, with a numerical value of 0.120–0.614 and a mean average of 0.328. The overall level of coupling and coordination was relatively low, suggesting a state of imbalance. A benign interactive coupling development was not formed, and there was still large room for improvement in coordination development.

According to the CCD classification criteria (Table 2), Tonghe Street (in Baiyun District) and Sha Mian Street (in Yuexiu District) were classified as Moderate Coupling Coordination. In fact, reaching the current level of balance largely lies in long-term sustainable development. Both streets possessed relatively strategic planning and strict controls for urban development, combined with the friendly construction of open spaces, preventing the imbalance caused by excessive construction in the process of urban development. The CCD index of the Tianhe District exhibited the worst, indicating that there was a serious imbalance between supply and demand in the planning and construction of GSI in this district (Figure 5). In terms of extreme decoupling coordination, Tianhe District had the most corresponding streets, followed by Liwan and Yuexiu District. The CCD index of the Huangpu District exhibited the highest, which was consistent with its performance in the CD evaluation. In terms of spatial distribution, regions above the level of primary coupling coordination were mainly located in the peripheral parts of the central urban area, such as the northeast of Baiyun, the north of Huangpu, and a small number of streets scattered around. The regions of decoupling coordination were mainly distributed in the entirety of Tianhe, the middle of Haizhu, and the south of Baiyun.

In comparison with the spatial distributions of supply levels and CCDs, it was found that the areas with a high supply were situated in the periphery of the central urban area, with low values of CCDs. In contrast, the urban center showed a higher CCD value, even though its GSI supply level was insufficient. The higher coordination level was driven by the core regions’ conditions of the local social economy, while the lower supply status was mostly constrained by the physical conditions of urban environments with high density [38,50]. To improve the overall coordination levels of GSI supply and demand, attention should be paid not only to the inputs of GSI in the periphery of the city, such as Baiyun and Huangpu Districts, but also to the social economic development level of this particular region.

Integrated with the CD and CCD indexes, a quadrant analysis was conducted to identify the distribution relationship among different streets, from “neither coupling nor coordinated (the third quadrant)” to “coupling coordination (the first quadrant)” (Figure 6). Fifty-eight streets were associated with the third quadrant, accounting for more than half of the total, which was mainly distributed in the districts of Tianhe, Baiyun, Haizhu, and Yuexiu. In contrast, there were only four streets in Liwan, as a minimum. The streets in this area were therefore in urgent need of an incremental supply of GSIs. Through the nuclear density estimation and comparison (Figure 7) subject to the GSI facility allocation scheme of local planning (Sponge City Implementation Plan), it was found that plenty of proposed GSI projects had not been effectively allocated in the “neither coupling nor coordinated” area, especially in the peripheral regions where Baiyun and Huangpu were located. Therefore, necessary adjustments were highly recommended for prioritizing the streets in the third quadrant of the urban planning scheme.

The fourth quadrant was defined as the “coupling but uncoordinated” area, covering a total of 38 streets scattered in various urban districts. Although this area possessed a large amount of GSI supply, the coupling relationship between supply and demand still remained unbalanced, which might be ascribed to a couple of facts: (i) the spatial distribution of GSI supply in this area was relatively concentrated and failed to take full responsibility [62]; (ii) A single GSI based on nature-oriented solutions could not cope with extreme rainstorms effectively, resulting in the occurrence of urban flooding [63]. Moreover, nine healthy streets were classified as “coupling coordination,” and their construction mode served as a role model for how similar streets should be incorporated into GSI planning.

## 4. Discussion

### 4.1. Contributions of Supply–Demand Structural Analysis to Flooding Management

The implementation of flooding control measures should be carried out, such as reworking impervious surfaces, supplementing the green stormwater infrastructure, or upgrading urban drainage systems. Unfortunately, it is nearly impossible to completely rebuilt existing urban drain systems due to the massive cost required [55]. Lin et al. [64] reported that in Shenzhen, a highly developed city, the cost of a permeable pavement retrofit was estimated to be as high as $320/m^2^. Similarly, in Guangzhou, a city neighboring Shenzhen, the cost of a region-wide permeable pavement retrofit was expected to be enormous, which seemed unrealistic. In addition, with the scarcity of available land resources, it is tough to expand the scale of green stormwater infrastructure in Guangzhou [31]. Therefore, as far as the sensible and cost-effective approach is concerned, priority should be given to streets where supply and demand are uncoordinated [35,64].

However, great differentiation was often not observed in the spatial pattern of GSIs’ supply and demand through the supply–demand assessment methods directly based on the urban ecological areas, resulting in waste of resources [65,66,67]. Unlike other supply–demand measurement methods, the approaches developed in this study provided a novel pathway to identify the streets with imbalanced supply–demand. The lagging areas can be quickly distinguished according to the results of the coupled and coordinated degree indexes. As such, the targeted planning decisions can be made in a timely manner for regions with uncoordinated CCD according to the local situation [66].

### 4.2. Causes of Supply–Demand Imbalances and Improvement Measures

According to the results of the GSI supply and demand coupling and coordination measurement, most streets have unbalanced GSI supply and demand. Fifty-eight streets exhibit a state of neither coupling nor coordinated, while thirty-eight streets show a state of coupling but uncoordinated. Only nine streets exhibit a balanced supply and demand (i.e., coupling coordination status).

In terms of the coupling but uncoordinated type streets, this type of street is concentrated in the older but more economically developed urban areas. In other words, spatial distribution characteristics of GSIs can match the special distribution of local socio-economic development, transport location, and political strategy (i.e., the “Matthew effect” of urban GSI services is formed), thus limiting the implementation of balancing GSI supply and demand [68]. Taking Sha Mian Street in the Yuexiu District as an example, its long history of development, superior natural ecological base, and convenient transportation location still affect the demographic composition, community attributes, and variations in perceptions of local urban residents, which successively influence the supply and demand for numerous public service facilities (e.g., green stormwater infrastructure). This phenomenon may cause agglomerations of various facilities with different scales and functions, such as scenic open spaces and green parks, leading to a systematically high level of coordinated development in Sha Mian. Therefore, to solve this problem, socio-economic development in urban fringe areas should be emphasized while integrating public facilities, such as green areas, municipalities, and social welfare facilities, into urban and rural planning.

Moreover, some of these streets are located at the edge of cities with relatively high GSIs, presumably due to the fact that the concentrated GSIs cannot adequately fulfil the stormwater function. Typical examples are Zhanqian Street in Haizhu District and Yongping Street in Baiyun District, where were cut off from residential catchments with large ecological parks, and the stormwater functions cannot be fully functioned, resulting in frequent flooding in residential areas. Meanwhile, Wang et al. [6] and Chang et al. [62] argued that decentralized GSIs are more effective in reducing runoff compared to centralized GSIs; therefore, a decentralized approach should be considered for rational planning and deployment of green stormwater infrastructure. As far as developing countries are concerned, with the fast progress of urbanization, it has been reported that China’s core built-up area will expand by about 3.97%, with 10% GDP growth [69]. Therefore, strict measurements for spatial controls should also be implemented to curb the rate at which urban green spaces are encroached by building sites [64]. Furthermore, types of streets usually possess a relatively large percentage of impervious surfaces and a relatively complex internal spatial pattern. Although there were GSIs arranged within the streets, which were mainly distributed in the central areas, with a small percentage in the peripheral areas of the city (e.g., Yuangang Street in Tianhe District and Shijing Street in Baiyun District), the quantity and quality of GSIs were not yet sufficient to meet the requirements of the study area. Therefore, enforcement action should be taken to improve the quality and service capacity of existing GSIs. Specific measures may include upgrading the park environment, renovating ecological greenways, and dredging river channels [70]. In other respects, reference can also be made for the improvements in the central city of Zhengzhou in streets with a large amount of impermeable surfaces. The rational design of road space and pavement is desired, using rainwater gutters and permeable paving materials, which can increase road space and pavement. Particularly, the adoption of rainwater gutters and permeable paving materials can not only increase permeability and rainwater storage but also increase planting space [68].

In addition, there are also some streets, such as Dongjiao Street in Liwan District and Dasha Street in Huangpu District, which have a better quantity and quality of GSI. However, other infrastructures were relatively poor (e.g., gray infrastructure), and they were not able to quickly discharge runoff during extreme storm events. In other words, A single GSI based on nature-oriented solutions could not cope with extreme rainstorms effectively, resulting in the occurrence of urban flooding [63]. Therefore, in addition to exploring nature-based solutions, structural measures for flood prevention, such as upgrading urban drainage systems, need to be strengthened to reduce the probability of flooding [71,72,73].

With respect to the neither coupling nor uncoordinated streets, a large part of this type of street is concentrated in the Tianhe, Baiyun, northern Haizhu, and southern Huangpu districts. Typical examples are Shahe Street in Tianhe District and Fengyang Street in Haizhu District, where a large number of urban villages have been established as a result of previous uncontrolled development, with a lack of pre-planning. These areas have massive impermeable surfaces, little green space, and a relatively poor living environment. Moreover, these streets attract a large number of groups with low incomes due to low housing rents. Without any improvement, significant damage may occur in these streets during the event of urban flooding. As such, measures are necessary to be taken to strengthen stormwater management strategies for urban flooding control. Lianhe street in Huangpu is a typical representative type for coupling and coordination streets, as it has undergone a radical urban village renovation, with strategies to increase vertical greenery and green roofs in an urban village that lacks open space, in addition to maintaining existing green spaces and upgrading the drainage system. leading to a decrease in urban flooding. Therefore, for this type of street, which is neither coupling nor coordinated, Lianhe street can provide a reference for improving the supply and demand of the GSI. Similarly, Shenyang, an old industrial base in northeast China, is also undergoing rapid urbanization and substantial renewal of old areas. Neither coupling nor uncoordinated streets could learn from the practice of vigorously promoting the maintenance and construction of private green spaces by residents and offering incentives [38].

Furthermore, a large number of streets in the central city of Guangzhou, which are in urgent need of improvement, have been neglected by the GSI facility allocation scheme for local planning (Sponge City Implementation Plan). In this regard, policymakers should focus more on streets, where GSIs supply and demand are uncoordinated, as assessed by our integrated method.

### 4.3. Limitations and Future Research Prospects

This study uses area and/or volume as an index of supply and demand to identify the streets where supply and demand are imbalanced. However, there still lacks a systematic analysis of the spatial interactions between supply and demand for GSI. In addition, GSI has been identified as a multifunctional and ecological infrastructure of cities. As part of future research, it will be pertinent to investigate GSI supply–demand relationships from broader perspectives (e.g., socio-economic as well as environmental). Moreover, assessing the supply–demand relationships associated with coordinated construction of GSIs and other types of infrastructures (e.g., gray pipe network system, road traffic system, open space system, etc.) will also shed light on better understanding of the appropriate allocation of urban infrastructure. Similarly, the hydrological model can effectively quantify the flood regulation ability of the GSI. This will give us the opportunity to accurately assess GSI supply and demand levels at the community scale, further analyze the causes of imbalances, and identify potential areas for improvement.

Moreover, given the fact that the distribution of GSI and the construction of sponge cities are different from city to city, the strategy for the GSI supply and demand optimization, which was derived from using only the central city of Guangzhou as a case study, may not be applicable to every city. In the future, research investigating GSI supply and demand levels in multiple cities to identify supply and demand allocation options should be carried out using cluster analysis, such as K-means or complex networks, for the further development of scientifically sound GSI planning.

## 5. Conclusions

This study developed an evaluation method for the spatial analysis of GSI supply and demand levels. Taking the central urban area of Guangzhou as an example, the coupling and coordination degrees of GSI supply, as well as the demand and supply–demand relationship, were identified in the street units of the city. Furthermore, the areas in urgent need of GSI incremental construction were ascertained. In addition, it was revealed that despite the continuous construction of GSIs in recent years, there still remained a large gap between the ideal supply and demand levels of GSI, suggesting that the construction of Sponge City was still at the initial stage. The findings of this study indicated that the extensive amount of investments in GSI was not only ineffective in improving the quality of urban hydrology but also susceptible to causing a waste of resource allocation. Policymakers need to prioritize areas where supply and demand are uncoordinated and take targeted measures tailored to local conditions.

## Figures and Tables

**Figure 1 ijerph-19-14742-f001:**
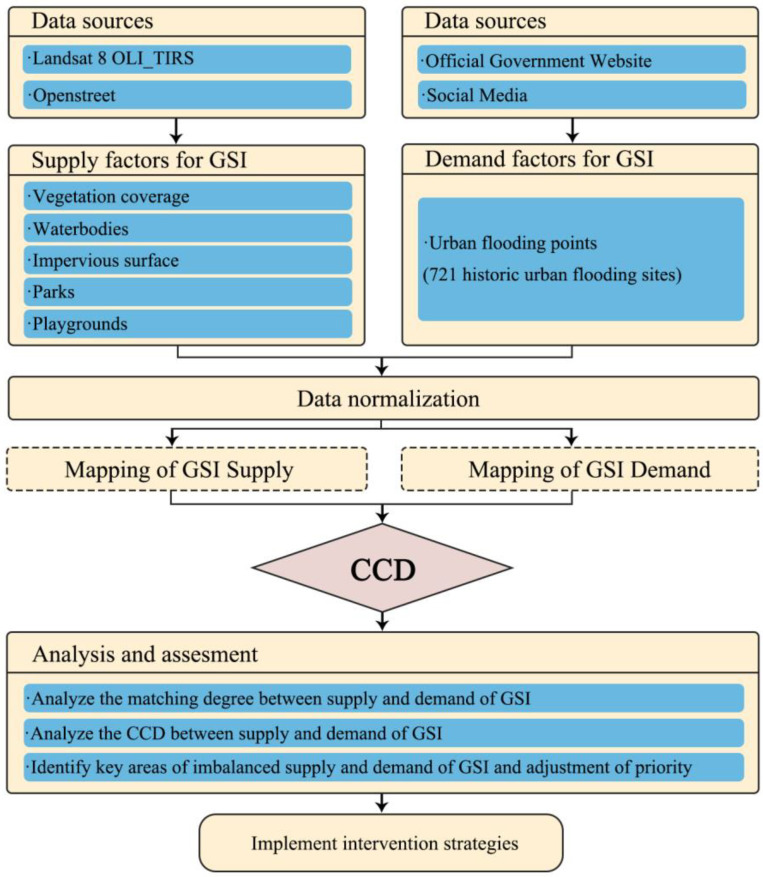
Study framework. Note: CCD—Coupling Coordination Degree model; GSI—Green Stormwater Infrastructure.

**Figure 2 ijerph-19-14742-f002:**
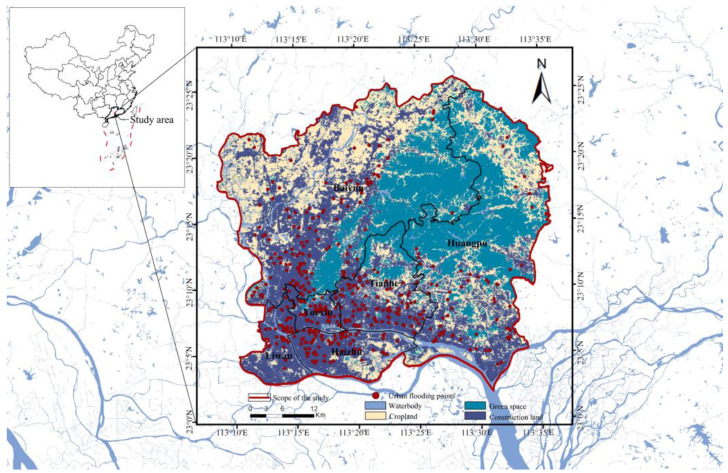
Study area.

**Figure 3 ijerph-19-14742-f003:**
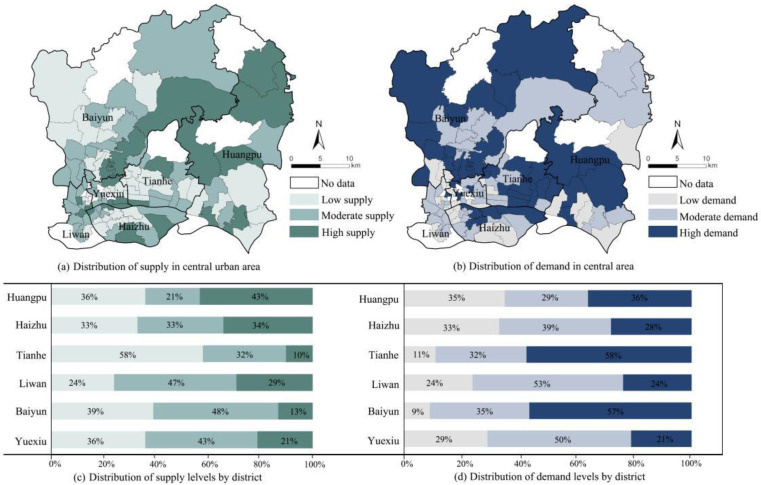
Spatial distributions of GSI supply and demand.

**Figure 4 ijerph-19-14742-f004:**
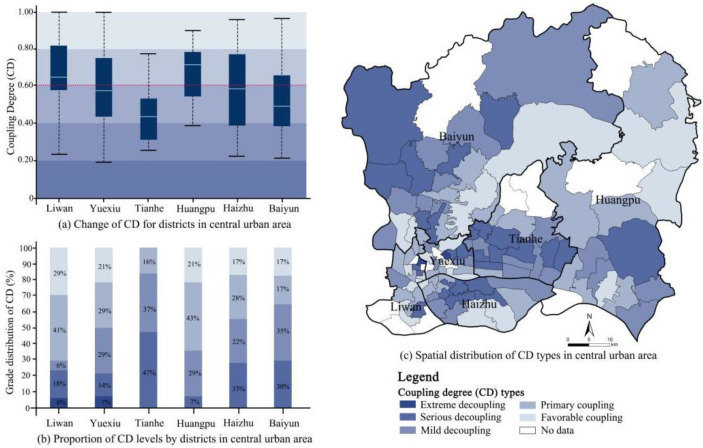
Mapping of Coupling Degree (CD) in the central urban districts.

**Figure 5 ijerph-19-14742-f005:**
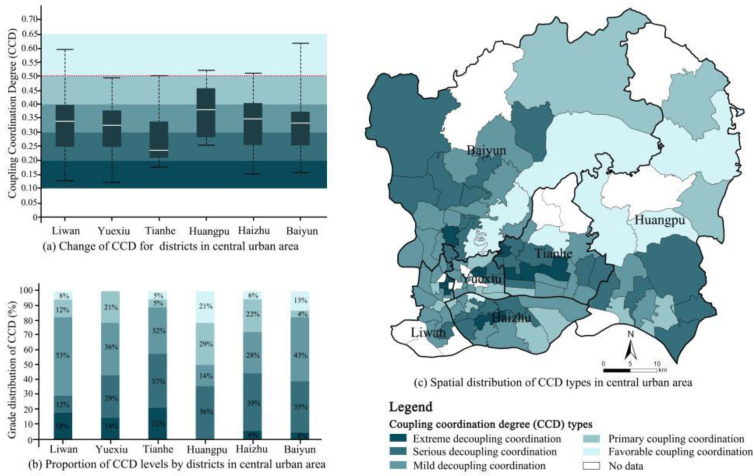
Mapping of the Coupling Coordination Degree (CCD) in central urban districts.

**Figure 6 ijerph-19-14742-f006:**
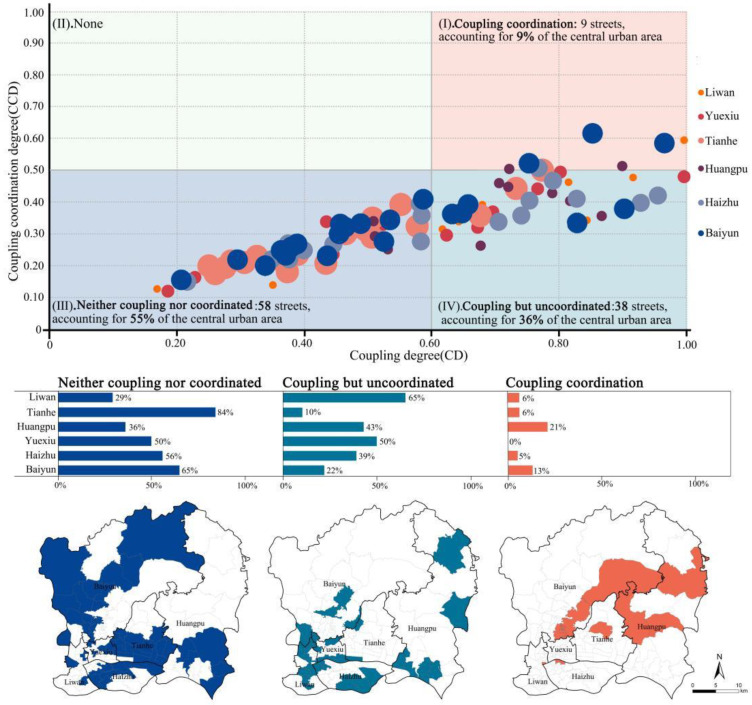
Quadrant analysis of CD and CCD.

**Figure 7 ijerph-19-14742-f007:**
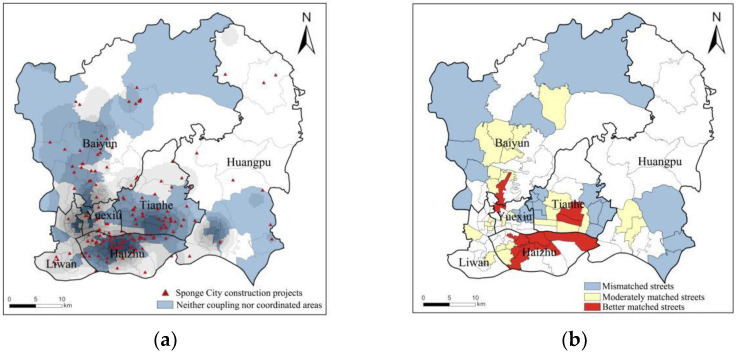
(**a**) Nuclear density analysis for Sponge City construction projects; (**b**) Results of matching CCD dysfunctional areas of Sponge City construction projects.

**Table 1 ijerph-19-14742-t001:** The index system for coupling coordination evaluation in Green Stormwater Infrastructure (GSI).

Systems	Subsystems	Format	Expected Impact	Data Sources
The demand of the GSI	Urban flooding points	Shapefile	Positive	Water Resources Department of Guangdong Province, China (http://swj.gz.gov.cn/, accessed on 15 December 2021) TouTiao (https://www.toutiao.com, accessed on 6 January 2022)
The supply of the GSI	Vegetation coverag	Shapefile	Positive	OpenStreetMap (https://www.openhistoricalmap.org/, accessed on 19 December 2021)
	Waterbodies	Shapefile	Positive	OpenStreetMap (https://www.openhistoricalmap.org/, accessed on 19 December 2021)
	Parks	Shapefile	Positive	OpenStreetMap (https://www.openhistoricalmap.org/, accessed on 19 December 2021)
	Playgrounds	Shapefile	Positive	OpenStreetMap (https://www.openhistoricalmap.org/, accessed on 19 December 2021)
	Impervious surface	Shapefile	Negetive	Landsat 8 Operational Land Imager_Thermal Infrared Sensor

**Table 2 ijerph-19-14742-t002:** Classification of the coupling coordination evaluation levels.

Coupling Index Value	Coupling Types	Coupling Index Value	Coupling Coordination Types
0~0.200	Extreme decoupling	0~0.200	Extreme decoupling coordination
0.200~0.400	Serious decoupling	0.200~0.300	Serious decoupling coordination
0.400~0.600	Mild decoupling	0.300~0.400	Mild decoupling coordination
0.600~0.800	Primary coupling	0.400~0.500	Primary coupling coordination
0.800~1.000	Favorable coupling	0.500~0.700	Favorable coupling coordination

**Table 3 ijerph-19-14742-t003:** Supply and demand composite value of Green Stormwater Infrastructure.

District	Demand Value	Supply Value	D-Value
Haizhu	0.725	0.264	0.461
Tianhe	0.804	0.221	0.583
Liwan	0.658	0.271	0.387
Yuexiu	0.716	0.259	0.457
Baiyun	0.794	0.266	0.528
Huangpu	0.722	0.301	0.421
Mean	0.737	0.264	0.473

## Data Availability

The data presented in this study are available on request from the corresponding author. The data are not publicly available due to part of them are being used in other studies that have not yet been publicly published.

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
