# Peer review of "Supply–Demand Evaluation of Green Stormwater Infrastructure (GSI) Based on the Model of Coupling Coordination"

_ijerph, 2022, doi:10.3390/ijerph192214742_

Round 1
Reviewer 1 Report
Overall the manuscript presents interesting information. More results than just one example in one city, however, would be more beneficial. Perhaps a summary table of other cities of other sizes could be included. While the results are interesting, it is important to know if the results are typical. For example, are results the same in other cities of this size and other sizes. Also, extensive revisions are necessary to correct grammar and spelling mistakes. It was difficult to understand the article in places but I couldn't tell if the reason was grammar issues only or more thorough explanations were needed. Additional comments include:
Line 68. What is an urban waterlog? The should be defined and/or discussed as I don’t believe it is a common term in all parts of the world.
Lines 71 and 73. What kind of facility are you referring to?
Line 74. Give an example of unreasonable spatial allocation. This may only take a few sentences to do.
Line 76. What is the meaning of “elasticity” here?
Line 90. I don’t understand the meaning of the sentence that begins with “However, the above methods…” Please revise and clarify.
Line 94. What is the “single dimension” to which you are referring?
Line 106. Is GI any different than GSI? If so, explain. If not, be consistent and use only one of the terms throughout the paper.
Line 111. What was the result of the Philadelphia study? Briefly summarize.
Line 118. What is the “critical coordination interval?”
Line 187. Provide a reference for the dispersion normalization method.
Line 212. What is Natural Breaks and/or why is it capitalized?
Line 298. How was the coupling relationship verified to be benign?
Line 346. Spell out numbers when they begin a sentence.
Line 348. Spell out numbers less than 10.
Line 377. Where these estimated costs or costs of existing and completed projects?
Section 4.3. Wouldn’t another limitation be that only one city was investigated/presented as an example? More data from other locations would be very informative but without this kind of additional data/results, it should be mentioned that other cities and countries may have different results.
Line 68. What is an urban waterlog? The should be defined and/or discussed as I don’t believe it is a common term in all parts of the world.
Lines 71 and 73. What kind of facility are you referring to?
Line 74. Give an example of unreasonable spatial allocation. This may only take a few sentences to do.
Line 76. What is the meaning of “elasticity” here?
Line 90. I don’t understand the meaning of the sentence that begins with “However, the above methods…” Please revise and clarify.
Line 94. What is the “single dimension” to which you are referring?
Line 106. Is GI any different than GSI? If so, explain. If not, be consistent and use only one of the terms throughout the paper.
Line 111. What was the result of the Philadelphia study? Briefly summarize.
Line 118. What is the “critical coordination interval?”
Line 187. Provide a reference for the dispersion normalization method.
Line 212. What is Natural Breaks and/or why is it capitalized?
Line 298. How was the coupling relationship verified to be benign?
Line 346. Spell out numbers when they begin a sentence.
Line 348. Spell out numbers less than 10.
Line 377. Where these estimated costs or costs of existing and completed projects?
Section 4.3. Wouldn’t another limitation be that only one city was investigated/presented as an example? More data from other locations would be very informative but without this kind of additional data/results, it should be mentioned that other cities and countries may have different results.
Reviewer 2 Report
I think this manuscript tackles an important question in the world of green stormwater infrastructure. I do think that this article can be published, but it the current form can be improved in certain key areas.
1) I think the introduction needs to more make the connecting between GSI and CD/CCD methodology more clearly. Admittedly, I am not that familiar with CD and CCD methodology but, even so, it is not abundantly clear how and why applying it to supply and demand of GSI is worthwhile, effective, and reasonable. Most of the introduction does a good job of setting up the rest of the manuscript, but the transition to CD and CCD methodology could be improved which will lead to a more coherent manuscript.
2) I could use a bit more explanation in the introduction about what "unreasonable spatial allocation" and "waterlogs" are. I have a good idea of what is meant by these terms, but some readers could benefit from specific explanations for each term.
3) Due to being unfamiliar with the methodology, I will not make specific recommendations towards the methods section, but hope that other reviewers can provide a more critical review of the methodology used in this manuscript.
4) I think the manuscript could be improved by expanding on the broad takeaway from the conclusions and introduce them in the discussion section. This will allow the authors to discuss more generalized recommendations about the implementation of GSI in China and beyond. I think the discussion in its current iteration is too focused on the specific region and could be improved by having more broad analysis of GSI implementation more generally.
5) I would also welcome some more critical hypotheses on why the shown gap between supply and demand occurs especially focusing on contrasting study areas where you have shown that it is not currently occurring.
Round 2
Reviewer 1 Report
Thank you for your careful and thoughtful attention to the issues raised in the first review. I believe the paper will be ready for publication after one minor revision.
The revision is on Line 78. I believe the word "insufficient" should be "sufficient."
Reviewer 2 Report
I would like to thank the authors for their attentiveness to the comments in the previous reviews. I believe the manuscript to be ready for publication following one minor revision:
~ Throughout the manuscript you have changed "waterlogging" to urban flooding which I think is a good change, but the change is not reflected in all the figures where the term "waterlogging" is still used.
